# Use of Magnetic Flux Leakage to Diagnose Damage to a Lift Guide Rails System with Reference to the Sustainability Aspect

**Paweł Lonkwic [1,*] , Tomasz Krakowski [2] and Hubert Ruta [2]**

[1] The Institute of Technical Sciences and Aviation, The University College of Applied Sciences in Chelm, Pocztowa Street 54, 22-100 Chełm, Poland
[2] Department of Machinery Engineering and Transport, Faculty of Mechanical Engineering and Robotics, AGH University of Krakow, al. A. Mickiewicza 30, 30-059 Krakow, Poland; tomasz.krakowski@agh.edu.pl (T.K.); hubert.ruta@agh.edu.pl (H.R.)
* Correspondence: plonkwic@panschelm.edu.pl

**Abstract:** The scientific objective of the conducted experimental research was to find an answer as to whether the application of magnetic flux leakage would be an effective tool for assessing the technical condition of lift guide rails in which the loss of thickness of the guide part is damage resulting from the brake whose operation destructively affects the surface shown. In particular, the scientific objective was to investigate the potential of this method in the context of quantitative assessment of the degree of damage featuring small increments in depth at the level of tenths of a millimetre. The conducted research was also aimed at determining the correlation of the effect of damage type with the nature of the signal recorded. The article presents the results of our own research, obtained from experiments on the use of magnetic flux leakage (MFL) to diagnose damage occurring on lift guide rails. During operation, lift guide rails are exposed to contact with the friction elements of brakes, resulting in the violation of their surfaces. Damage to the working surfaces of guide rails increases the vibration of the device, noise and wear of other components of the lift, such as guide rails. Currently, diagnostics of lift guide rails are not carried out, and their replacement depends on their technical condition. However, from an economic point of view, there are situations wherein their condition allows their use without their replacement with new ones. This was the main factor that guided the authors; we used a diagnostic head of our own design for the tests. The obtained measurement results showed that magnetic flux leakage can be used with great success to diagnose damage to guide rails. The results obtained in the laboratory shall be further developed in the form of research on correlating the signals obtained from magnetic sensors and the size of the damage, which shall eventually allow for a final quantitative assessment of guide rails regarding their technical condition. The conducted research fits into the scope of sustainable development by reducing the need for the consumption of electric energy and the emission of harmful substances into the atmosphere in the overall production balance. This will be made possible by implementing the developed head in industrial practice in the context of assessing the need to replace guide rails with new ones. The economic and environmental efficiency that is the basis of sustainable development in the context of lifts can be achieved at the modernisation stage by repeated (further) use of as many components as possible. An example of this is the guide rail system, the reuse of which is possible after a prior assessment of its wear and tear.

**Keywords:** passenger lifts; diagnostics; non-destructive testing; ropes; magnetic field; magnetic flux leakage (MFL); sustainability; ecology

## 1. Introduction

Passenger lifts, as technical objects, belong to a narrow group of transportation devices called material handling equipment. Their main purpose is to transport people or goods between floors of a building. Based on the method of transmission, the lifts are divided into two categories:

- Friction-powered lifts;
- Hydraulically powered lifts.

Diagnostics of damage at the operational stage is an important factor affecting the further operation and safety of technical objects. Due to difficulties related to the testing of technical objects in real conditions and transferring such conditions to the laboratory, research on the topic of this paper has not been widely reported on in the literature.

However, considering the development of lifts, research on the presented subject matter will develop, and its results will be necessary for the development of methods that allow monitoring of various lift components, which in the future will allow faster responses to emerging disturbances in the operation of these devices. Currently, the use of the magnetic field in its broad sense is described with reference to various research fields in scientific publications. However, there are no research papers or related scientific publications describing the application of non-destructive testing methods, in particular, the magnetic method for diagnostics and support in assessing the technical condition of lift guide rail system elements. The selected publications testifying to the wide applicability of research methods based on magnetic phenomena with reference to various types of objects and physical phenomena are presented below.

Dogonchi et al. [1] described their own research work related to induced heat transfer with the involvement of natural convection, in a rectangular enclosure with a corrugated circular heat source in the form of a radiator, influenced by the application of magnetic phenomena and nanoparticles. In publication [2], the behaviour of fluid in a cavity enclosure is analysed, which is an important issue in the domain of fluid mechanics. In this paper, a hybrid nanofluid composed of water and ethylene glycol (50–50%) is evaluated as the base fluid, which contains hybrid $MoS_2$-$TiO_2$ nanoparticles, in an octagon with an elliptical cavity in the centre. The subject of the analysis was the effects of the radiation parameter, porosity and magnetic parameter on the temperature distribution and flow lines of fluid as well as on the local and average values of the Nusselt number. As a novelty, the Taguchi method was applied to the design of the research. The obtained results showed that along with an increase in the Rayleigh number from 10 to 100, the average Nusselt number improved by about 61.82%. In addition, in terms of correlation, the Rayleigh number had the highest contribution compared to other factors in the obtained equation, by about 61.88%.

Izadi et al. [3] conducted a computational analysis of thermal gravitational convection in a porous chamber under the influence of an inclined periodic magnetic force. The equations formulated using the single-phase nanofluid approach, the extended Brinkman–Darcy model for transport processes in the porous layer with the local thermal disequilibrium approach and the Boussinesq approximation for the buoyancy force and magnetic field description were solved by means of the Galerkin finite element method. The subject of the analysis was the effect of Darcy, Hartmann and Rayleigh numbers, magnetic field periodicity, magnetic field angle, thermal conductivity coefficient and porosity of the medium on the flow pattern and thermal characteristics. Based on the results obtained, it was found that the periodic magnetic field parameters had a non-monotonic effect on the heat transfer performance. In publication [4], there is a detailed description of a magnetic head design, as well as initial results of measurements made with this head. The dimensional variety of the lift guide rails used requires the versatility of diagnostic equipment. The designed head provides the possibility of adjusting its dimensions relevant to the measurement procedure. Universality was achieved by the application of inserts adapted to the dimensions of the guide rail under test. The guide rails can be tested without being disassembled. Furthermore, the tests conducted under laboratory conditions confirmed that the developed solution is metrologically correct. The proposed solution and methodology are part of the development of new diagnostic methods and tools.

In the current state of knowledge, studies on the application of the wavelet analysis method in building diagnostic expert systems for magnetic applications are known. One example is the research presented in articles [5,6], in which the authors described a system

for data acquisition (measuring/testing head—recorder—software) and data analysis in magnetic testing of the technical condition of steel T-T straps. The analysis referred to the evaluation of signals recorded during testing, and its main determinant was the selection of an appropriate type of wavelet for filtering the diagnostic signal and further analysis. A special role in the whole system was that of the expert software to support decision-making processes. This software required absolute determination of the correlation between wear processes in fatigue testing of T-T straps and the results of magnetic testing in order to develop the final criteria for acceptance or rejection of this T-T strap structure. In publication [7], the use of a pioneering concept of the magnetic head is described for assessing the technical condition of lift guide rails that are the running track of lift equipment. The preliminary tests were performed on the test stand designed for this purpose. The correct operation of the developed measuring head was verified on specially prepared flat bars with holes. The results obtained during the laboratory tests showed that the proposed solution of the head concept can be applied to measurements in real conditions. Article [8] presents the process of optimisation of the prototype head magnetic core for magnetic diagnostics of steel–polyurethane T-T straps.

The initial version of the prototype head, after being verified in situ on a real object (crane/lift) and proving its effectiveness, was subjected to further development. This process consisted of optimising the head design while keeping in mind metrological and performance objectives such as the dimensions and weight of the unit. This optimisation concerned the magnetic circuit, which is the main element of the head, to determine the objectives mentioned. In order to achieve the first objective, i.e., better performance of the diagnostic signal relative to noise, it was necessary to numerically determine the topography of the magnetic field in the magnetic circuit under test, the T-T strap and the surrounding, for the predetermined boundary conditions. A number of design scenarios were analysed using DoE tools in the ANSYS environment. For various design scenarios, the input data were changed, such as the following:

- Magnetic properties of structural materials including steel for magnetic core elements and field source materials (permanent magnets);
- Dimensions and shapes of individual structural elements of the magnetic core;
- Mutual relations between magnetic core elements.

The numerical results of magnetic field topography were compared for one of the design variants to the results of real measurements of magnetic induction in the same selected characteristic areas, proving the convergence of the numerical experiment and its multiple scenarios with reality. Conclusions from the analysed results made it possible to develop guidelines for changing the elements forming the magnetic core. The selected additional head elements made of non-ferromagnetic materials were replaced with FDM plastic printed elements for greater weight reduction of the unit. The final determination of the degree of design improvement was proven through comparative studies of the initially developed prototype and the version after optimisation. These tests were conducted under laboratory conditions on a steel and polyurethane T-T strap with precisely modelled damage. The test results obtained and their statistical characteristics were subjected to a detailed analysis.

Recalling the studies that would be more coincident with those described in this article, it is required to mention the previously published studies of the authors. Lonkwic et al. [9], using their previous experience in the magnetic diagnostics of ropes and T-T straps and the design of heads for these applications, and based on the similarity of physical issues, described the possibility of applying the magnetic method to diagnose the lift guide rails. Similarly, as for the T-T straps, it is necessary to properly magnetise the structure under test, where the magnetic field is disturbed in the area of discontinuities on the working surface from the guide rails. The disturbance in the distribution of individual components of the magnetic field is registered by appropriate magnetic sensors. The recorded measurement results and their analysis allowed further work in this domain, which will contribute to the development of methods for the diagnosis of T-T straps used in various technical facilities.

Publication [10] includes the analysis of a single-walled composite nanoshell (SWCNS) exposed in a critical torsional stability situation. With reference to the magnetic field acting on nanostructures of small size, a three-dimensional magnetic field was evaluated, which included magnetic effects along the circumferential, radial and axial coordinate system. The obtained results of the small-scale strain gradient model and the first-order shear deformation shell theory (FSDST) were compared with those from other publications. It was observed that the transverse component of the magnetic field was the determining factor for the torsional stability of the shell.

Article [11] presents the comparison of the utility properties of mine lift wire ropes determined by visual, magnetic and strength tests. There are correlations between the indices that are the values characterizing particular types of tests. Article [12] presents the results of an analysis of steel wire ropes installed on oil rigs. The mentioned studies were the result of existing regulations and the occurrence of an emergency situation for the rig crew. The tests were carried out using the NDT method, the purpose of which was to verify the technical condition of all wire ropes on the rig. The test results were compared with the guidelines found in the existing standards for the use of wire ropes on oil rigs. Publication [13] provides a study on the possibility of using the passive magnetic method to diagnose steel ropes which are a structural component of building structures. The characteristics obtained using magnetic methods allowed for a quantitative assessment of damaged wires in the rope. The multidimensional damage detection experiments were conducted using self-magnetic flux leakage (SMFL) for different states of rupture of parallel bundles of steel wires. Based on this, a quantitative evaluation method with a Boltzmann curve approach was proposed, which is a new approach to this type of issue.

The use of magnetic fields is also described in publications on other topics, proving that researchers in various fields of engineering are interested in this phenomenon. Publication [14] presents the results of the effect of make-up water subjected to a constant magnetic field (B = 1 T) on the properties of concrete. The fillers such as coal fly ash (CFA), phosphogypsum (PB) and native potato starch dispersed in epoxy resin [EP(S)] were used to produce Portland cement (PC) based on concrete. The use of magnetised make-up water had a beneficial effect on the properties of concrete by reducing its water absorption and increasing its flexural and compressive strength. In publication [15], based on the model calculations, there is an analysis of the effect of rail system geometry on the distribution of magnetic induction, and the field generation efficiency was compared. Also, the effect of materials surrounding the rails on the magnitude of magnetic field induction in the launcher is discussed. Theoretical calculations were verified during the launcher model testing.

Paper [16] presents the results of research in medicine, where microrobots controlled by means of optical and magnetic fields were used. Their use was confirmed by experimental and numerical simulation analyses. The results introduce an innovative method of precise modulation, providing a new method for treating diseases based on gut bacteria. Publication [17] presents an analytical model of the motor that predicts the magnetic field distribution based on the Laplace and Poisson equations, which are solved using the variables separation method. Based on the model prepared, three connected permanent magnets were proposed, accounting for both edge-cutting and polar arc-cutting structures, which are chamfered, rounded and rectangular combinations. Assuming a constant amount of edge cutting, the electromagnetic characteristics of the three permanent magnet combinations are comparable to the simulation obtained using the finite element method. The obtained results of the mathematical and simulation model were confirmed by tests with a prototype motor under laboratory conditions. Article [5] describes research conducted on the possibility of applying wavelet analysis to evaluate the results of magnetic wire rope testing. In publications [18–21], the results of research on the possibility of localising wire rope defects on the basis of recording and analysing magnetic fields resulting from local changes in magnetic properties are presented. In publication [22], there is an attempt to correlate the number of bends in a wire rope with the magnetic induction value.

As can be seen from this review of publications, the use of magnetic flux leakage to assess the condition of technical objects or their components, due to its nature, is not widely applied directly to technical objects. This is mainly due to the fact that most components are made of ferromagnetic materials, which means that their evaluation using MFL requires their isolation from the system. On the other hand, such components as wire ropes, guide rails or others already isolated from the technical object are perfectly suitable for such an assessment. In this study, the possibility of applying magnetic flux leakage to evaluate guide rails under conditions close to real ones is pointed out. The results described in this paper are complementary to articles [8,10], which present the results of tests performed under laboratory conditions only, that considered damage done in the form of notches at a depth that far exceeded the damage resulting from the action of the brake on the guide rail surface. The attention is focused on the applicability of this method for visible and perceptible damage. Since in real conditions, the surface of the guide rail is contaminated with grease, the damage shown in Figure 12b is not visible to the naked eye due to the lubricant layer applied. Therefore, in order to assess the condition of guide rails, it was necessary to find the answer to whether the magnetic flux leakage is able to indicate the occurrence of the mentioned damage, which was the scientific objective of this study.

## 2. Characteristics of the Test Subject

The subject of the study is damage (shown in Figure 3) to the S surface of the steel guide rail shown in Figure 1, which occurs during operation due to its interaction with the friction elements of brakes.

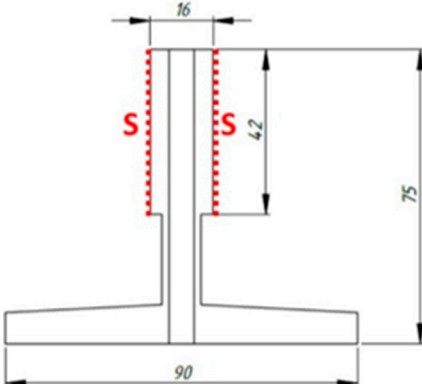

**Figure 1.** Cross section of rail T90/A drawing, all dimensions in millimetres [10].

Since each lift travels along two guide rails positioned opposite each other (Figure 2), any plastic damage to their surfaces causes an increase in vibration, noise and wear of the cabin frame guide rail elements.

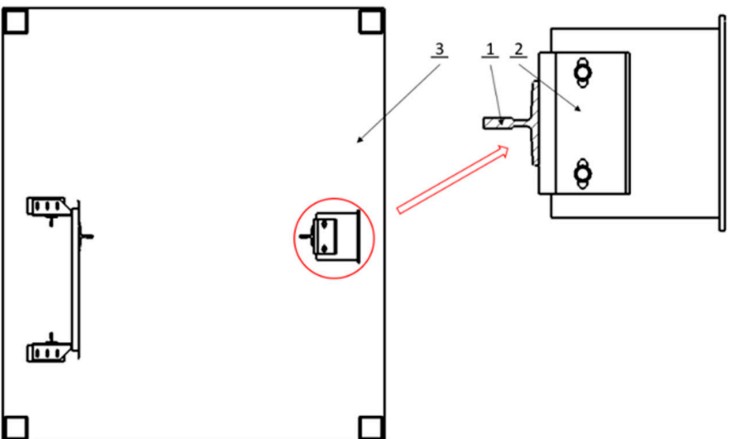

**Figure 2.** Location of guide rails in the lift shaft: 1—guide rail, 2—bracket, 3—shaft.

The guide rail is made of E235B steel according to [22], and the physical and mechanical properties are shown in Table 1.

**Table 1.** Physical and mechanical properties of E235B material [23].

| | |
|---|---|
| Density $\gamma$ | 7.8 g/cm$^3$ |
| Young's modulus E | 210 GPa |
| Elongation $A_5$ | 23% at 20 °C |
| Poisson's ratio $\nu$ | 0.29 |
| Tensile strength $R_m$ | 405 MPa |
| Yield strength $R_e$ | 210 MPa |

The lift brakes can be activated by the following factors:

- Uncontrolled increase in nominal speed due to control system failure;
- Uncontrolled increase in nominal speed due to broken T-T straps.

Figure 3 shows an example of a guide rail with a damaged surface after operation of the brake. For the correct further operation of the lift, this surface must be levelled by mechanical methods, which result in the reduction of cross-sectional area at the location where the brake roller contacts the guide rail. For both their evaluation and monitoring of the technical condition of guide rail surfaces, the device (magnetic head) was developed whose operation is based on the analysis of magnetic flux leakage.

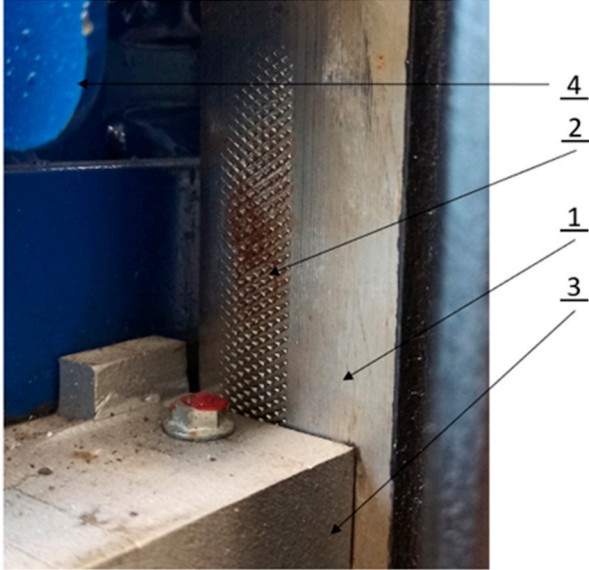

**Figure 3.** Example of guide rail surface damage due to the brake operation: 1—guide rail, 2—damage to the active surface, 3—brakes, 4—frame.

The research described in the article was carried out using the magnetic method that makes it possible to localise surface damage to the lift guide rails (loss of metallic section) on the basis of a symptom in the form of a change in the magnetic flux leakage. The research was conducted by applying the technical solution and method that was recognized by *Elevator World* magazine (Project of the Year 2020—the first prize in the Elevators—Modernisations and Repairs category) and was filed as a utility model with the Patent Office of the Republic of Poland (no. Ru072199m).

A simplified model of the designed and constructed measuring head, which was used in the laboratory tests that were carried out as described further, is shown in Figure 4. The basic elements of the head are as follows [5]:

1. The body to which individual components and elements are mounted; in the proposed solution, it has the function of a sliding system of guidance on the working surface of the guide rail.
2. The rotary encoder, together with a disc mounted on the encoder shaft, which is in contact with the surface of the tested guide tail, allows correlation of the recorded signal of the magnetic sensor with the path travelled by the measuring system (this allows the localisation of the defect along the tested guide rail length). The position adjustment depending on the tested guide tail thickness and the assurance of appropriate pressure are possible thanks to a spring and mounting on a rotary axis in a socket located in the body.
3. Permanent magnets (2 pieces), oppositely polarised, which are the source of the magnetic field of the circuit.
4. The sensor and its displacement system, depending on the dimensions of the guide rail to be tested; it is possible to move the sensor in two perpendicular directions marked A and B in Figure 4.
5. The magnetic jumper, the element that closes the magnetic circuit; due to the coherence forces of magnets, the jumper does not require additional mounting.
6. Magnet covers that protect magnets from uncontrolled displacement.

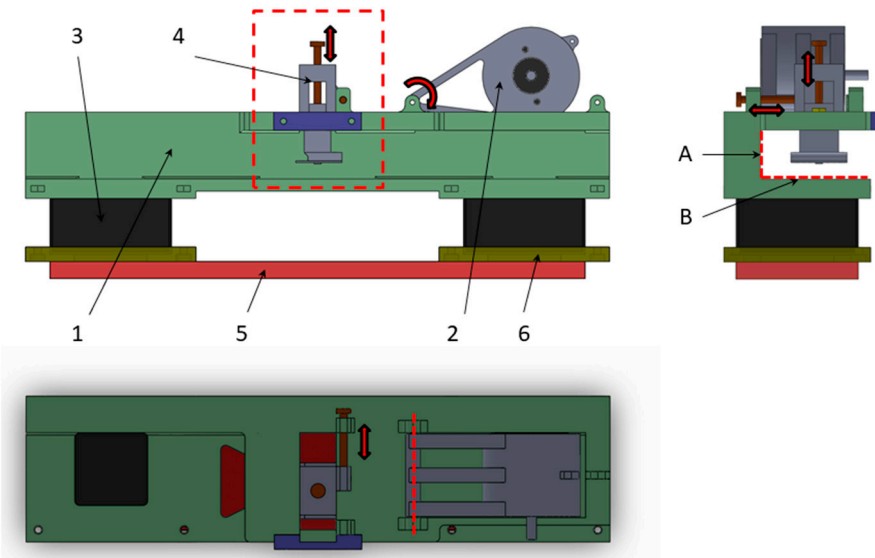

**Figure 4.** CAD 3D model of the measuring head [5]: 1—measuring head body, 2—encoder, 3—permanent magnet, 4—sensor, 5—magnetic jumper, 6—cover plates, A—adjusting plane, B—adjusting plane.

To verify the assumptions made and ultimately confirm the applicability of magnetic flux leakage for fault detection in the diagnostic case being discussed, the numerical analyses of magnetic field distributions were carried out for the CAD model of the proposed measuring head magnetic core and a section of the guide rail with a defect (Figure 5).

The numerical experiment was carried out for boundary conditions, taking into account material parameters such as, but not limited to, the BH primary magnetisation curve of SA1020 steel defined for the magnetic head jumper and the steel of the guide rail, the BH demagnetisation curve of the NdFeB—N52 rare earth material (Br = 1450 mT, Hc = 800 kA/m) defined for permanent magnets and their polarisation direction, as well as the magnetic permeability of the surrounding air domain, which is the natural measurement environment in which magnetic flux leakage is generated. The mathematical and physical instruments used in the analyses based on the FEM and Maxwell's Equations (1)–(4) implemented in ANSYS have been described in detail in item [10].

$$\nabla \times \{H\} = \text{rot}\{H\} = \{J\} + (\partial D / \partial t) \tag{1}$$

$$\nabla \times \{E\} = rot\{E\} = -(\partial B / \partial t) \tag{2}$$

$$\nabla \cdot \{B\} = div\{B\} = 0 \tag{3}$$

$$\nabla \cdot \{D\} = div\{D\} = \rho \tag{4}$$

where

{H} = magnetic field intensity vector [A/m];

{J} = electric current density vector [A/m$^2$];

{D} = electric field induction vector [C/m$^2$];

t = time [s];

{E} = electric field intensity vector [V/m];

{B} = magnetic field induction vector [T] or [Wb/m$^2$];

$\rho$ = total density of electric load [C/m$^3$].

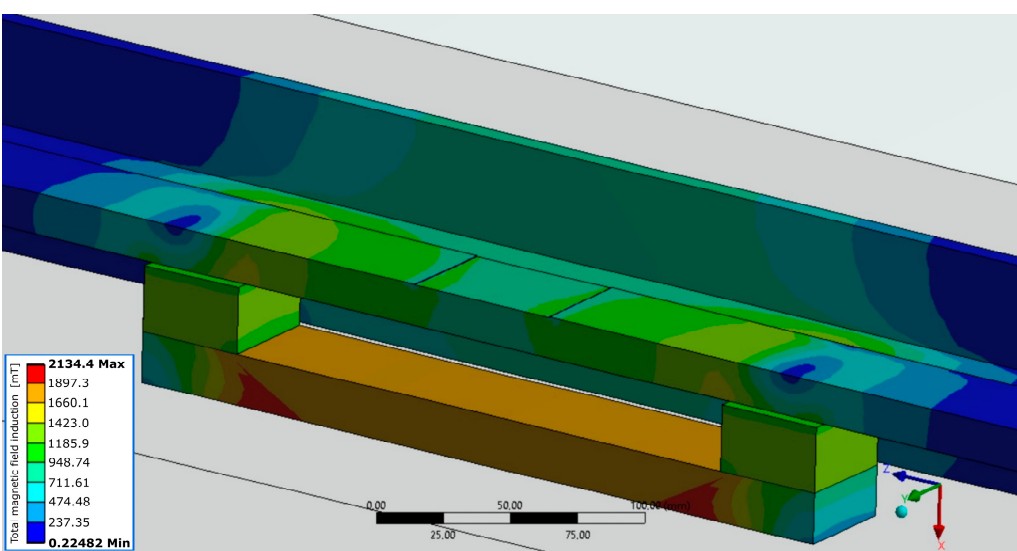

**Figure 5.** Distribution of magnetic field induction in the magnetic core of the measuring head and the guide rail with a defect.

In the model analysed, the damage was shaped in the form of a 0.3 mm deep and 60 mm long notch in the lateral surface of the guide rail head along which the lift guide rails move when the cabin is in motion. The damage is continuous in nature with sharp edges (Figure 5).

To better visualise the changes in magnetic flux leakage, a detection line was defined just above the lateral surface of the guide rail head (Figure 6) in the damaged area. This line reflects the motion trajectory of the sensor installed in the magnetic head in question. A total of 200 values of the total magnetic field induction (Figure 6) as well as the B$_x$ (Figure 7) and B$_z$ (Figure 8) components of this induction were taken uniformly from the line.

The Bx and Bz components, presented in Figures 7 and 8, of magnetic flux leakage induction on the detection line along the guide rail just above the defect clearly indicate the field disturbance at the beginning and end of the defect of a continuous nature. In addition to the changes in induction values in the graphs, the same FEA results are presented in the background in the coloured graphical form from the detection line observed perpendicular to the lateral surface of the guide rail head. In this experiment, the beginning and end of the damage were modelled as sharp transitions and, due to this, the changes in the numerical signal are intense. In the case of a smooth transition of the undamaged area of the guide rail into the damaged one and vice versa, the signal changes may be of a milder nature; nevertheless, the numerical analyses confirmed that the assumed concept was right and preliminarily validated it, which allowed us to proceed with making the prototype and conducting laboratory measurements.

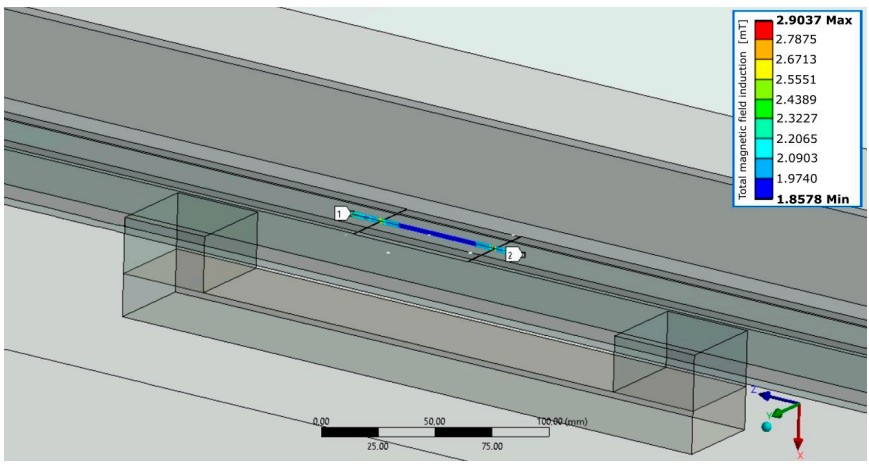

**Figure 6.** Distribution of magnetic flux leakage induction on the detection line over the fault.

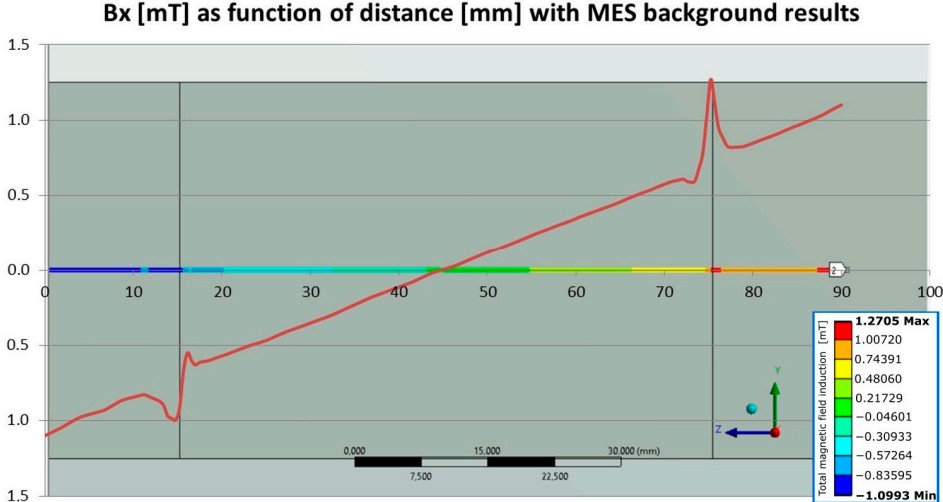

**Figure 7.** Changes in the Bx component of magnetic flux leakage induction on the detection line along the guide rail just above the defect—perpendicular view to the lateral surface of the guide rail head.

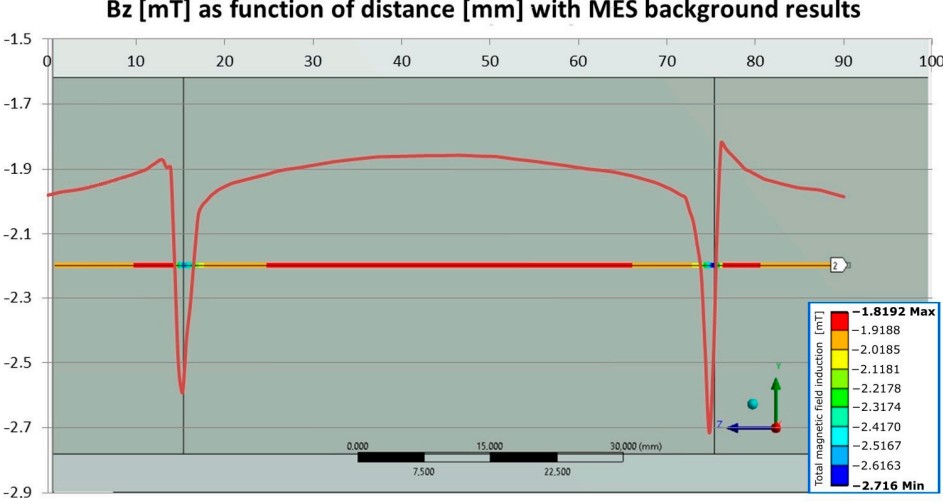

**Figure 8.** Changes in the Bz component of the magnetic flux leakage induction on the detection line along the guide rail just above the defect—perpendicular view to the lateral surface of the guide rail head.

All non-magnetic components of the head were made using 3D-FDM (Fused Deposition Modelling) printing technology. Polylactide (PLA) was used as the main material for printing the body, while stainless steel screws and bolts were chosen for the fasteners. This technology and material were chosen due to the strength aspect, i.e., the absence of additional external mechanical loading on the system, apart from cohesion forces of the magnets, and the technological aspect, i.e., the ease and speed of making subsequent prototypes of the head. The head used for the laboratory tests is shown in Figure 9 [5].

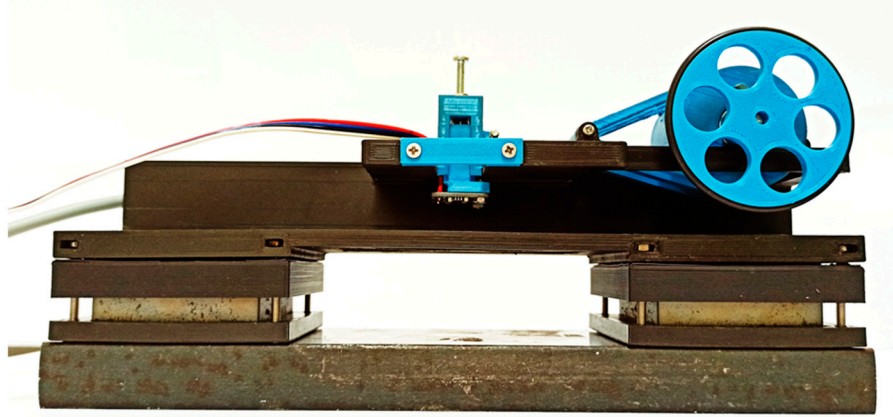

**Figure 9.** Magnetic head used in laboratory tests [5].

Laboratory tests of the guide rail section with modelled damages are shown in Figure 10. The obtained test results and practical experience will enable the development of the next version of the measuring head with functional modifications introduced. Conducting the in situ tests will also require the system's modification in terms of data acquisition.

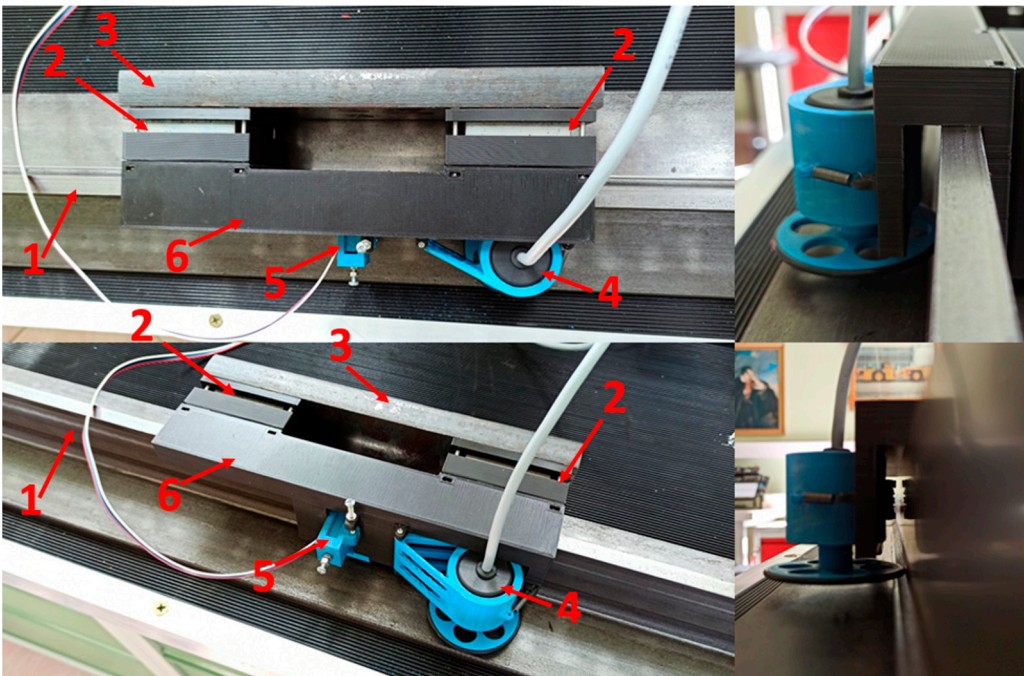

**Figure 10.** Laboratory tests of the lift guide rail section [5]: 1—guide rail under test, 2—permanent magnet, 3—magnetic jumper, 4—encoder, 5—sensor with its attachment, 6—measuring head body.

The measurement system shown in Figure 11 consisted of the recording module of magnetic flux leakage disturbance and the displacement module. Recording, synchronisation and recording of signals from both modules were carried out by a program developed

for the purpose of measurements (LabView). The recording module of magnetic flux leakage change consisted of the digital 3-axis MLX90393 magnetometer and the ArduinoMega microcontroller. The displacement module consisted of the ER30 inclement encoder and the NI USB-6216 measurement card [5]. The test was conducted under laboratory conditions. Stable movement of the head at a constant speed of about 0.05 m/s was ensured by moving it manually relative to a stationary guide rail. The measurement sensor used is the MLX90393 3-axis Hall-effect sensor with the measurement range of up to 5000 uT and the resolution of 0.5 uT. The selected resolution of the magnetic field transducer is appropriate for the mentioned application, taking into account the fact that the existing defects generate changes in the amplitude of magnetic flux leakage in the range of about 50–100 uT. This allows for a reasonably accurate representation of diagnostic signal changes. The use of a Hall-effect sensor made the recorded signal independent of any instability and changes in the speed of movement of the head relative to the guide rail, which is a significant drawback of the inductive sensors often used in similar applications. The sampling rate was 20 ms, thus giving the recording frequency of 50 Hz. For the test speed adopted, this gives a measurement every 1 mm along the length of a guide rail. The applied encoder and roller arrangement allowed the displacement signal to be recorded with a resolution of no more than 0.6 mm.

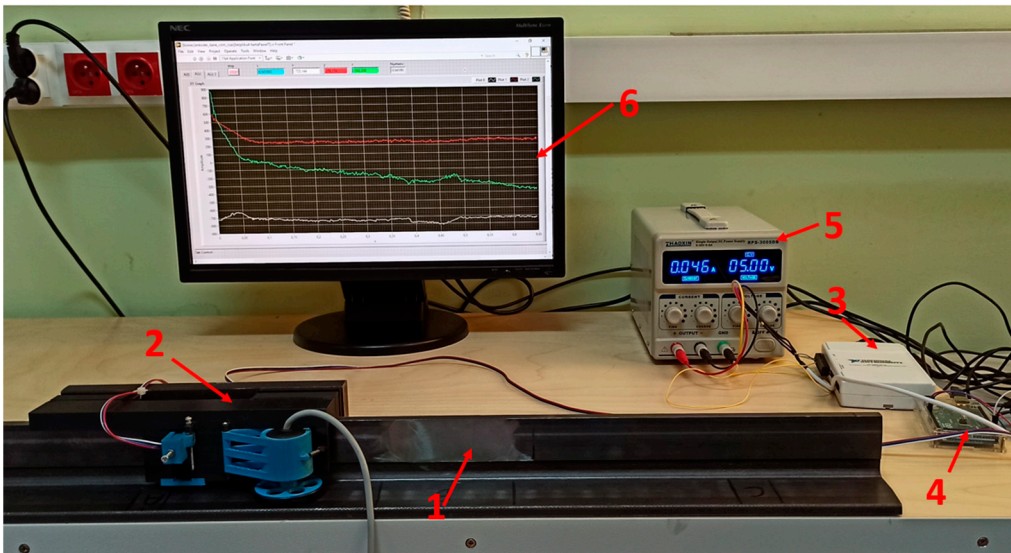

**Figure 11.** Measurement system used for testing under laboratory conditions [5]: 1—guide rail under test, 2—measuring head, 3—NI USB-6216 measurement card, 4—ArduinoMega microcontroller, 5—laboratory power supply, 6—data logger software.

### 3. Measurement Results and Their Analysis

The laboratory tests were carried out on a guide rail on the surface of which mechanical damage was made to simulate a change in cross section.

The first type of damage was of a step-like nature and is marked with the letter A in Figure 12, while the second type was of a continuous nature and is marked with the letter B. The continuous damage, which corresponds, by its type, to the damage made on the surface of guide rails after grinding at the point of engagement of safety stops, was done in two grinding stages. The first stage consisted of light sanding of the surface (hereafter marked as B1), and in the second stage, the continuous sanding was deepened in the same area (hereafter marked as B2). The point-like damage remained unchanged throughout the testing process. The measurements of magnetic flux leakage on the guide rail surface were carried out after each grinding stage, for each variant of the damage.

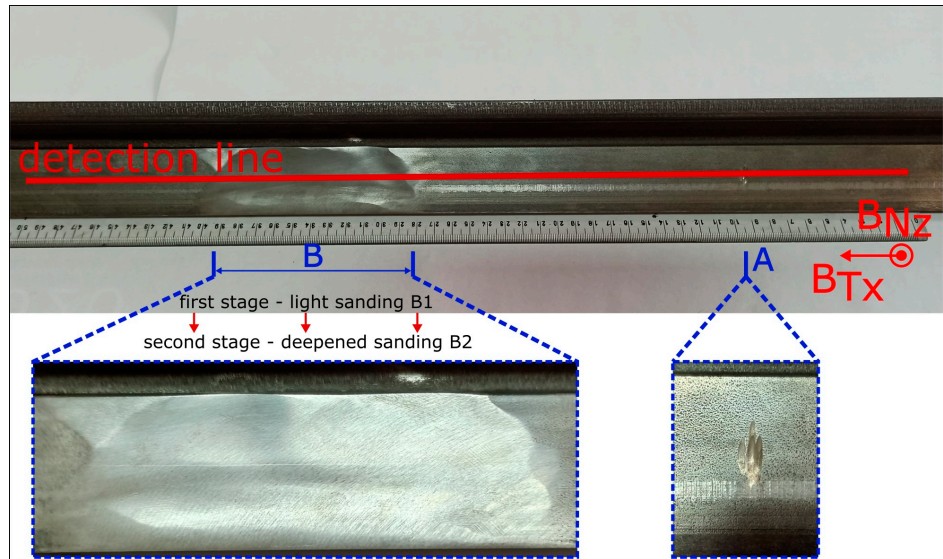

**Figure 12.** Modelled damage on the lateral surface of guide rail head (A—step-like damage, B—continuous damage, $B_{Tx}$—tangential component along the guide rail length, $B_{Nz}$—normal component).

In order to synchronise and record data from the magnetometer and the path transducer, an interface was developed with a program in the LabView environment for the tests conducted. The measurements were carried out with the same magnetisation direction and the same position of the sensor and the measurement line along which the sensor moved. Two components of magnetic induction over the tested guide rail surface were measured, the tangential component along the guide rail length $B_{Tx}$ and the normal component $B_{Nz}$. Figure 13 shows the values of normal component $B_{Tx}$ and tangential component $B_{Nz}$ of the magnetic flux leakage induction recorded along the measurement line for two variants of the guide rail damage (the first, B1, and the second, B2, grinding stages).

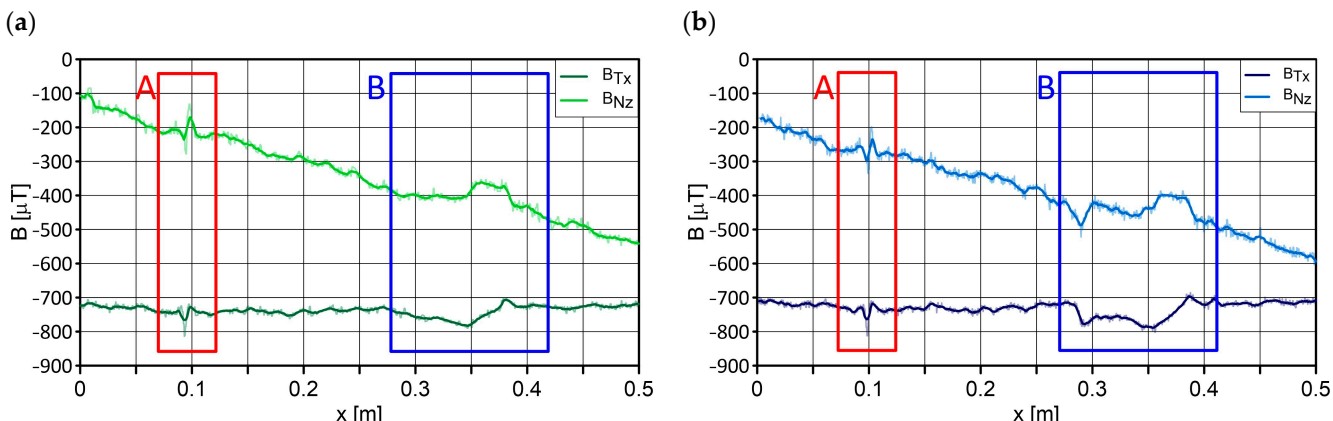

**Figure 13.** Recorded values of the normal component $B_{Tx}$ and the tangential component $B_{Nz}$ of magnetic induction for two variants of damage: (**a**)—point-like damage A and continuous damage B1—first stage of grinding, (**b**)—point-like damage A and continuous damage B2—second stage of grinding—deepened continuous damage.

The gradient of tangential $gradB_{Tx}$ and normal $gradB_{Nz}$ components for the corresponding magnetic flux leakage component was determined as the absolute value of a derivative of the mean recorded magnetic induction components (5) and (6). Also, the resultant gradient $gradB_{TxNz}$, calculated as the product of the gradients of tangential $gradB_{Tx}$ and normal $gradB_{Nz}$ (7) components, was determined.

$$gradB_{Tx} = \left| \frac{B_{Tx(n)} - B_{Tx(n-1)}}{\Delta x} \right| \tag{5}$$

$$gradB_{Nz} = \left| \frac{B_{Nz(n)} - B_{Nz(n-1)}}{\Delta x} \right| \tag{6}$$

$$gradB_{TxNz} = gradB_{Tx} \cdot gradB_{Nz} = \left| \frac{B_{Tx(n)} - B_{Tx(n-1)}}{\Delta x} \right| \cdot \left| \frac{B_{Nz(n)} - B_{Nz(n-1)}}{\Delta x} \right| \tag{7}$$

where

$B_{Tx(n)} - B_{Tx(n-1)}$—difference between adjacent mean values of the tangential component of magnetic induction;

$B_{Tx(n)} - B_{Tx(n-1)}$—difference between adjacent mean values of normal component of magnetic induction;

$\Delta x$—distance between adjacent measurement points.

The distribution of gradients of the tangential component $gradB_{Tx}$ and the normal component $gradB_{Nz}$ are shown in Figures 14 and 15. The gradient of the product $gradB_{TxNz}$ is shown in Figure 16. In the distribution of damage gradients in the first variant, the stage after the first grinding (B1), two peaks are visible, which correspond to the focused damage and the final part of the continuous damage. In the distribution of gradients in the second variant of damage, the stage after deepening the continuous damage (B2), three maxima are visible, which correspond successively to the focused damage and the beginning and end of the continuous damage.

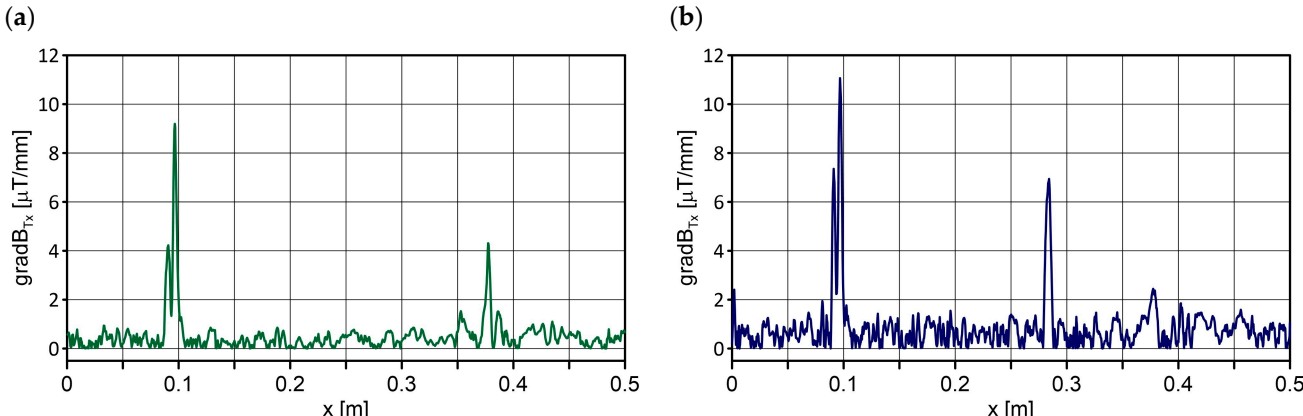

**Figure 14.** Gradient of the tangential component gradB$_{Tx}$ for two variants of damage: (**a**)—point damage A and continuous damage B1—first grinding stage, (**b**)—point damage A and continuous damage B2—second grinding stage—deepened continuous damage.

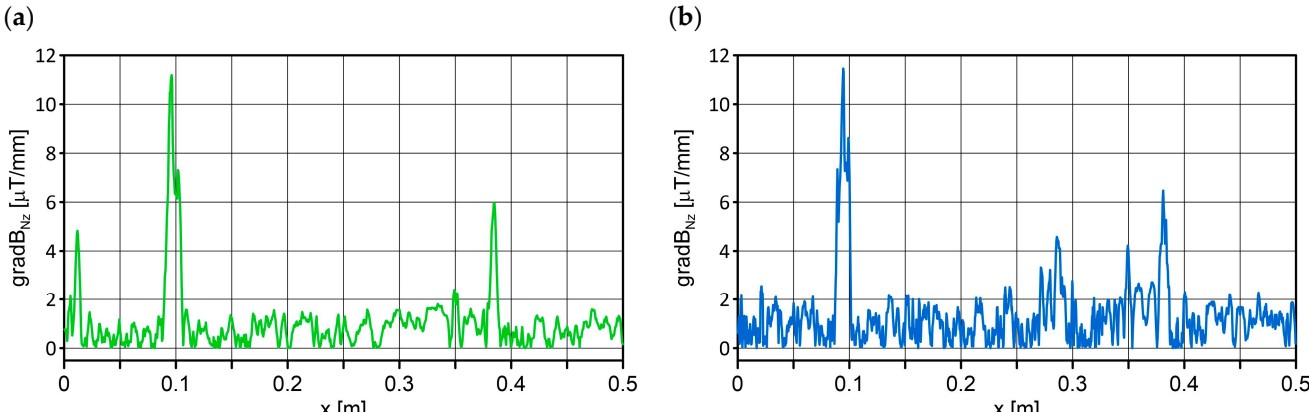

**Figure 15.** Gradient of the normal component gradB$_{Nz}$ for two variants of damage: (**a**) point damage A and continuous damage B1—first grinding stage, (**b**) point damage A and continuous damage B2—second grinding stage—deepened continuous damage.

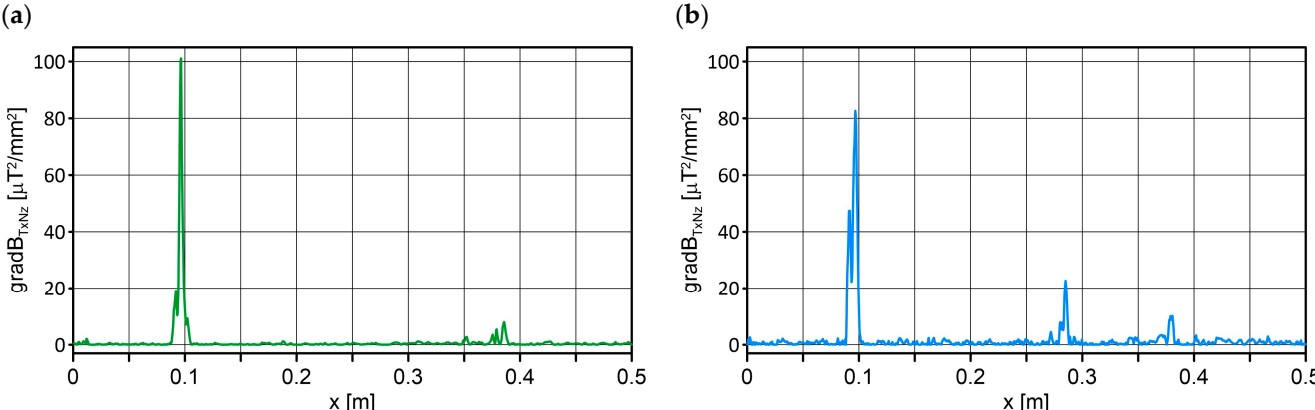

**Figure 16.** Gradient of the product gradB$_{TxNz}$ for two damage variants: (**a**) point damage A and continuous damage B1—first grinding stage, (**b**) point damage A and continuous damage B2—second grinding stage—deepened continuous damage.

## 4. Discussion on Results Obtained

The tests carried out under laboratory conditions confirmed the established thesis. The obtained measurement characteristics in Figures 14–16 clearly indicate that the magnetic flux leakage is an excellent medium for analysing the technical condition of lift guide rails. The damage shown in Figure 12 was made relative to the beginning of the guide rail section at a distance of 100 mm—damage A, and at a distance of 280 to 390 mm—damage B, which was reflected in the characteristics obtained. As the cavity (damage to the guide rail surface) increases, the magnetic induction distribution makes it possible to diagnose its occurrence. In Figure 13a, for cavity B1 obtained during the first grinding stage, the change in magnetic induction is less visible than for cavity B2 in the area from 0.28 to 0.39 m, which is correlated with the damage made.

The product gradient characteristics *gradB$_{TxNz}$*, shown in Figure 15b, indicate that damage that is "point-like" but deep is well visible in the form of a single peak, while the blended surface having a certain length is less visible, with a smaller cavity, as shown in Figure 15a, versus a cavity with a larger value, as shown in Figures 15a and 17, where the first signal peak corresponds to the beginning of the damage and the second one to its end.

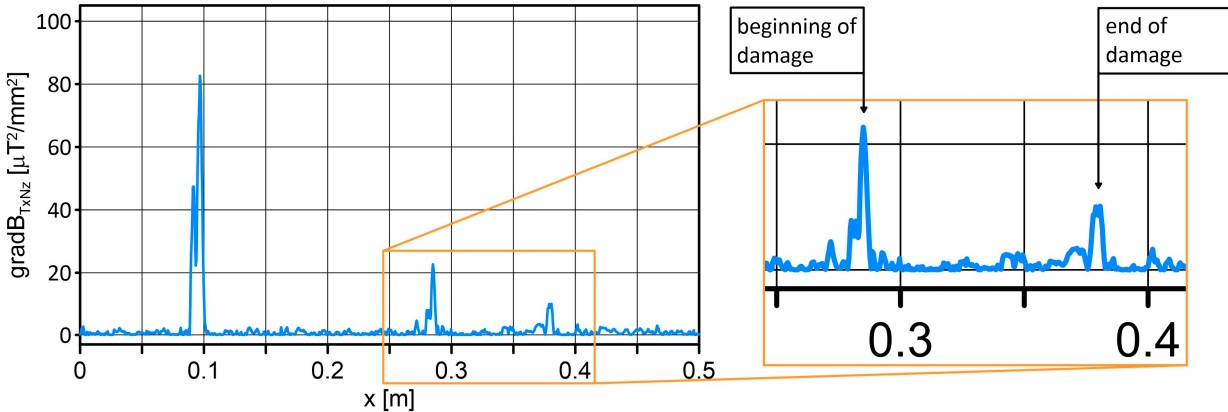

**Figure 17.** Damage area detected on the guide rail surface as the resultant gradient *gradB$_{TxNz}$*.

## 5. Conclusions

The assumptions for the simulations carried out with the use of the FEM were confirmed by the results of experiments accomplished on a real object. The FEM simulation as a step towards the scientific goal was to confirm the adopted methodology, the initial dimensions of the equipment and its metrological properties depending mainly on the magnetic properties of the magnetic core. In the numerical experiment, the diagnostic

signal variation (variation of the magnetic leakage field) caused by the local occurrence of a defect was obtained at a level of about 0.7 mT or 700 uT, while in the practical experiment, the variation reached about 50–100 uT. This difference is mainly due to three reasons. The first reason is that the defect of the "B" nature (Figure 12) in the numerical model has a sharp change in geometry (geometric notch), while in reality, this grinding is of a mild nature, generating smaller changes in the magnetic leakage field. The second reason is that during the practical experiment, magnets with a smaller height than in the FEM model were used, having slightly lower magnetic energy, due to the large cohesion (attraction) forces between the head and the guide rail. The excessive coherence forces that occur in practice hinder stable relative motion between the test head and the guide rail. A lower magnetic energy in the magnetic core that the guide rail in the test is a part of also leads to smaller magnetic leakage field changes originating from the defect. The third reason is that in the case of the FEM analysis, sampling was carried out continuously, which in the case of sharp changes in geometry allowed for the accurate representation of rapid changes in the magnetic leakage field and the associated diagnostic signal. When sampling every 2 mm in a real experiment, even in the case of sharp/abrupt changes in geometry, it is possible to unintentionally miss a localised maximum from a change in the diagnostic signal and record a sample below the maximum. Despite this, the magnetisation level of the guide rail and the high resolution of the sensor used allowed the effective detection and localisation of damage sites, both of a point and continuous nature. The method of analysing the diagnostic signal adopted in this article, based on the use of gradients between the recorded samples from two components of the magnetic leakage field responding to disturbances from defects, and the multiplication of these gradients, provides additional amplification of the signal in relation to the unavoidable noise, resulting, among other things, from vibration during the measurement. The key to the developed equipment's effectiveness in the presented application is the selection of appropriate magnetic properties of the magnetic core, which was enabled by the authors' know-how, the selection of adequate measuring transducers for the level of an expected change in the diagnostic signal and the effective mathematical analysis of results. Taking into account the arguments presented and the scientific goal achieved, the research carried out and its application, confirmed by its ability to detect and localise defects, are of an innovative nature. In addition, regarding global conditions referring to sustainable development, the presented solution fits into these assumptions. The application of the described method will eliminate cases wherein the replacement of guide rails is performed unnecessarily due to their good technical condition, which will translate into the reduced consumption of electric energy and the reduced emission of toxic components resulting from production and road transport into the atmosphere.

Thus, the laboratory tests carried out allow us to draw the final conclusions:

- As can be seen from the characteristics obtained, the signal change is affected by the damage depth. The greater it is, the more evident it is in the form of changes on the graph. The developed equipment has high potential for assessing the level of damage.
- Magnetic flux leakage can be applied to localise damages and to diagnose the technical condition of lift guide rails with great success.
- The efficiency of the method itself in laboratory conditions is satisfactory, which leads us to believe that after refinement of the head design, it can be applied in real conditions.
- The developed head, due to its research nature, needs to be refined in terms of design.

## 6. Directions for Further Research

Taking into account the experience and the conclusions of the research and development work carried out so far, further research should be carried out aimed at, among other things, the following:

- Further research should be conducted to confirm or deny the use of this method to assess the depth of damage with the required accuracy level of about 0.1 mm;

- Determination of the effect of magnets and magnetic circuit parameters on coherence forces relative to the guide rail to facilitate mutual motion during the test;
- Development of solutions to reduce vibration during the measurement and noise levels in the diagnostic signal;
- Development of software solutions to enable automatic real-time data analysis based on the proposed or similar mathematical operations, combined with an audio and/or visual indicator to inform the user of acceptable or unacceptable object status (green LED/red LED) to support the decision-making process;
- Verification of the equipment's effectiveness when the guide rails are covered with the grease with iron chips that may appear in service;
- Optimisation of the design in terms of metrology (better signal quality) and functionality (ergonomics of assembly, disassembly, movement, data recording and acquisition).
- Improvement of the measurement resolution along the guide rail length, e.g., from 2 mm to 0.5 mm, by increasing the sampling rate while maintaining the same effective test speed.

**Author Contributions:** Conceptualisation, validation, investigation, H.R. and T.K.; writing—original draft preparation, H.R., T.K. and P.L.; writing—review and editing, H.R., T.K. and P.L.; visualisation, H.R., T.K. and P.L. All authors have read and agreed to the published version of the manuscript.

**Funding:** This research received no external funding.

**Data Availability Statement:** The data presented in this study are available on request from the corresponding author.

**Conflicts of Interest:** The authors declare no conflicts of interest.

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
