# Peer review of "Use of Magnetic Flux Leakage to Diagnose Damage to a Lift Guide Rails System with Reference to the Sustainability Aspect"

_sustainability, doi:10.3390/su16051980_

Round 1
Reviewer 1 Report
Comments and Suggestions for Authors
Dear Authors,
the manuscript entitled "Use of stray magnetic field to diagnose damage to the lift guide rails system" by Pawel Lonkwic and co-authors deals with simulation and laboratory tests of damages that may occur during use of elevator components. The Authors used the stray magnetic field as a diagnostic parameter in assessing the condition of elements made of E235B material. The article presents the practical application of the magnetic method, which is its undoubted advantage. I appreciate the contribution that the Authors made in simulation, experimental testing as well as preparing the manuscript. However, in my opinion the manuscript needs to be significantly improved in some fields and some general remarks as well as the specific comments are bellow.
Evaluation of the paper, general remarks, editorial comments/typos:
1) What is the type of article (Article, Review, Communication, etc.)? Authors should detail the type of article based on the information it contains. What is "[email protected]" in line 9?
2) I found neither in the abstract nor in the whole main text what is the scientific purpose of the article. In scientific papers, the purpose should be clearly indicated, which is then achieved by the results of the research and its analysis.
3) Abstract section should present quantitative results and not only the most important qualitative results and/or generic considerations. Significant improvements are expected in this section of the manuscript.
4) line 125 - there is "in the article 12" and should be in the article [12].
5) At the end of paragraph I (Introduction), the novelty of the proposed solution and the scientific goal should be defined in detail. The article should also include a justification for the knowledge gap it fills. In its current form, the authors did not included the main goal of presented research. There is some research on stray magnetic field, so the authors must clearly indicate what they have done beyond what is already published in the literature. The differences between the submitted article in Sustainability journal and references [8] and [10] in which the Authors discussed similar topics should be indicated in detail.
6) Chapter 1 requires significant modifications. Each description of scientific publications begins with "In the article...". The description of current research results and applications of stray magnetic field should be understandable and user-friendly (it should be one story, discussing detailed aspects of the topic related to the article). The provided description does not meet these requirements. Moreover, the literature analysis is very poor - 14 references, 5 of which are the Authors of the reviewed work (36%). Please modify chapter 1, add more publications related to the basics of the method, its application in other technical facilities, especially as diagnostic methods.
7) The text in chapter 1 and other chapters is not justified, e.g. lines 43-50 and others.
8) Please also include information (preferably with pictures documentation) what typical damage occurs in the guide rail system.
9) line 161 - thete is "3 - shaft" and should be 3 – shaft. The same remark for line 181.
10) Fig. 1., Fig. 2., Table 1 and others - the caption is not formatted according to the guidelines of the journal, please see the instruction for Authors.
11) line 212 - the desciption of the number placed in figure should be also in Figure caption.
12) line 243 - there is "0.3mm deep and 60mm" and should be 0.3 mm deep and 60 mm.
13) Fig. 10 and Fig 11 - please add marks of the most important research equipment and a description in the caption of the Figures.
14) Figure 12 - Where are B1 and B2 marked in the figure? Furthermore, please explain the difference between defect A shown in this figure and the defect placed in references [8]. In my opinion, it is the same picture (same defect), only reversed.
15) Authors present their results but without any discussion supported by the literature. When the results are not discussed and conveniently supported by the open literature, questionable conclusions are obtained. Currently, the article looks more like a report from simulation and test than a scientific article. What does it mean "poczatek uszkodzenia" in Figure 17?
16) One sentence presenting directions for further research is definitely not enough. Please expand this part of the article. It is best to supplement the manuscript with an additional Conclusion section (please divide the chapter 4 for two chapters), which will contain a summary and directions for further research.
17) Please read the instructions, how to describe the references at the end of the article in the Authors guide and change it. Currently, the references at the end of the text are not in line with the journal requirements. Journal title and volume should be written by italic font and year should be bold.
Taking into account the above comments, I believe the article in current form should be reject. I also believe that the topic presented in the article is relevant. I hope these suggestions can help to improve the quality of this paper. I encourage the Authors to improve the manuscript and resubmit to Sustainability journal.
I wish you all the best.
Reviewer
Comments on the Quality of English LanguageDear Authors,
The article uses the personal form in several places, e.g. line 65 "they found...". This is not correct in high-quality articles. It suggests modifying this part of the article. Please check the entire article in terms of personal form (line: 81, 99, 101, 137, 286, 416 and others).
In several places, the article contains complex sentences that are difficult for the reader to understand. Please read the entire text of the article and simplify the sentences.
Kind regards
Reviewer
Author Response
Dear Reviewer,
Thank you for your review of our article, which will significantly increase its attractiveness to the reader. The article has been read again and the expanded sentences have been changed.
1) What is the type of article (Article, Review, Communication, etc.)? Authors should detail the type of article based on the information it contains. What is "[email protected]" in line 9? – THE TYPE OF ARTICLE HAS BEEN CORRECTED. PROBABLY NOT ALL INFORMATION WAS LOADED AT THE STAGE OF SUBMITTING THE ARTICLE TO THE AUTHORS’ PANEL. A GENERAL MAIL RECORD WAS REMOVED FROM LINE 9.
2) I found neither in the abstract nor in the whole main text what is the scientific purpose of the article. In scientific papers, the purpose should be clearly indicated, which is then achieved by the results of the research and its analysis – THE INFORMATION ON THE SCIENTIFIC PURPOSE HAS BEEN INCLUDED IN THE INTRODUCTION.
3) Abstract section should present quantitative results and not only the most important qualitative results and/or generic considerations. Significant improvements are expected in this section of the manuscript.- The article has been completed and corrected
4) line 125 - there is "in the article 12" and should be in the article [12]. –THIS TEXT WAS NOT IN ITALIC IN THE SUBMITTED VERSION.
5) At the end of paragraph I (Introduction), the novelty of the proposed solution and the scientific goal should be defined in detail. The article should also include a justification for the knowledge gap it fills. In its current form, the authors did not included the main goal of presented research. There is some research on stray magnetic field, so the authors must clearly indicate what they have done beyond what is already published in the literature. The differences between the submitted article in Sustainability journal and references [8] and [10] in which the Authors discussed similar topics should be indicated in detail. – CHAPTER I HAS BEEN SUPPLEMENTED BY AN EXPLANATION OF THE DIFFERENCE BETWEEN ARTICLES 8 AND 10 AND THE CURRENT CHAPTER.
6) Chapter 1 requires significant modifications. Each description of scientific publications begins with "In the article...". The description of current research results and applications of stray magnetic field should be understandable and user-friendly (it should be one story, discussing detailed aspects of the topic related to the article). The provided description does not meet these requirements. Moreover, the literature analysis is very poor - 14 references, 5 of which are the Authors of the reviewed work (36%). Please modify chapter 1, add more publications related to the basics of the method, its application in other technical facilities, especially as diagnostic methods. – THE ARTICLE HAS BEEN EXPANDED TO INCLUDE OTHER PUBLICATIONS ON THE USE OF MAGNETIC FIELDS IN VARIOUS ASPECTS OF ENGINEERING.
7) The text in chapter 1 and other chapters is not justified, e.g. lines 43-50 and others. – The article has been completed and corrected.
8) Please also include information (preferably with pictures documentation) what typical damage occurs in the guide rail system. – The article has been completed and corrected.
9) line 161 - thete is "3 - shaft" and should be 3 – shaft. The same remark for line 181. –THIS TEXT WAS NOT IN ITALIC IN THE SUBMITTED VERSION.
10) Fig. 1., Fig. 2., Table 1 and others - the caption is not formatted according to the guidelines of the journal, please see the instruction for Authors. – The article has been completed and corrected.
11) line 212 - the desciption of the number placed in figure should be also in Figure caption. – THE DESCRIPTION OF DRAWING HAS BEEN COMPLETED.
12) line 243 - there is "0.3mm deep and 60mm" and should be 0.3 mm deep and 60 mm. –THIS TEXT WAS NOT IN ITALIC IN THE SUBMITTED VERSION.
13) Fig. 10 and Fig 11 - please add marks of the most important research equipment and a description in the caption of the Figures. -– THE DESCRIPTIONS OF DRAWINGS HAVE BEEN COMPLETED.
14) Figure 12 - Where are B1 and B2 marked in the figure? Furthermore, please explain the difference between defect A shown in this figure and the defect placed in references [8]. In my opinion, it is the same picture (same defect), only reversed. – THE DESIGNATIONS HAVE BEEN CHANGED IN THE TEXT.
15) Authors present their results but without any discussion supported by the literature. When the results are not discussed and conveniently supported by the open literature, questionable conclusions are obtained. Currently, the article looks more like a report from simulation and test than a scientific article. What does it mean "poczatek uszkodzenia" in Figure 17? – THIS EXPRESSION IN FIGURE 17 HAS BEEN TRANSLATED. THE RESULTS OBTAINED HAVE BEEN DISCUSSED IN MORE DETAIL.
16) One sentence presenting directions for further research is definitely not enough. Please expand this part of the article. It is best to supplement the manuscript with an additional Conclusion section (please divide the chapter 4 for two chapters), which will contain a summary and directions for further research. – The article has been completed and corrected
17) Please read the instructions, how to describe the references at the end of the article in the Authors guide and change it. Currently, the references at the end of the text are not in line with the journal requirements. Journal title and volume should be written by italic font and year should be bold. – THE LIST OF PUBLICATIONS HAS BEEN HARMONISED WITH THE JOURNAL REQUIREMENTS.

Reviewer 2 Report
Comments and Suggestions for Authors
Dear Authors, In the work entitled Application of a scattered magnetic field to diagnose damage to the elevator guide system. Research on the use of a scattered magnetic field to diagnose damage occurring in elevator guides is presented. Interesting research was carried out in the work, but before publication it should be corrected in the following points:
1. In lines 33 and 41 it is worth adding literature.
2. When discussing individual works, it is worth adding the name of the first author in the text, this will certainly improve the readability of the work. Sometimes it's not clear what the work is.
3. The introduction is described in detail. The articles were analyzed for their introduction to the topic. however, there are some errors, e.g. [6] is not given as a footnote but as plain text.
4. To make the article more readable, it can be divided into sections: results and discussion together and conclusions separately.
5. The discussion lacks references to literature and comparisons with other works. It's worth adding this to your work.
Author Response
Dear Reviewer,
Thank you for your review of our article, which will significantly increase its attractiveness to the reader. As a result of your review, we have made the following changes to the content of the article:
Dear Authors, In the work entitled Application of a scattered magnetic field to diagnose damage to the elevator guide system. Research on the use of a scattered magnetic field to diagnose damage occurring in elevator guides is presented. Interesting research was carried out in the work, but before publication it should be corrected in the following points:
- In lines 33 and 41 it is worth adding literature. – AS THE TEXT WAS WRITTEN BY US AND WAS NOT QUOTED, WE DO NOT USE THE QUOTATION HERE .
- When discussing individual works, it is worth adding the name of the first author in the text, this will certainly improve the readability of the work. Sometimes it's not clear what the work is. – THE ADDITION OF AUTHORS’ NAMES IS NOT REQUIRED IN THIS JOURNAL SO WE HAVE NOT INCLUDED THEM.
- The introduction is described in detail. The articles were analyzed for their introduction to the topic. however, there are some errors, e.g. [6] is not given as a footnote but as plain text. – THE FOOTNOTE HAS BEEN COMPLETED.
- To make the article more readable, it can be divided into sections: results and discussion together and conclusions separately. – The article has been completed and corrected.
- The discussion lacks references to literature and comparisons with other works. It's worth adding this to your work. – The article has been completed and corrected.

Reviewer 3 Report
Comments and Suggestions for Authors
The article presents a new method of diagnosing damage occurring on the lift guide rails by the use of stray magnetic field (MFL). But the manuscript shloud be improved.
1. There are many methods of damage detection. What are the current problems? What are the advantages of magnetic method?
2. The research content needs to be enriched. such as, the measurement accuracy, sensibility, reliability and speed should by explained.
3. Why the resultant gradient is the the product of the gradients of tangential component and normal component? Please explain.
Comments on the Quality of English LanguageThe Spelling and grammar errors need to be corrected.
Italics and standardized form need to be distinguished, especially formulas, parameters and units.
The Fig 16 and Fig 17 are same. The text of Fig 17 should be in English.
Author Response
Dear Reviewer,
Thank you for your review of our article, which will significantly increase its attractiveness to the reader. As a result of your review, we have made the following changes to the content of the article:
The article presents a new method of diagnosing damage occurring on the lift guide rails by the use of stray magnetic field (MFL). But the manuscript shloud be improved.
- There are many methods of damage detection. What are the current problems? What are the advantages of magnetic method? - The article has been completed and corrected.
- The research content needs to be enriched. such as, the measurement accuracy, sensibility, reliability and speed should by explained. - The article has been completed and corrected.
- Why the resultant gradient is the the product of the gradients of tangential component and normal component? Please explain. – The article has been completed and corrected.
Comments on the Quality of English Language
The Spelling and grammar errors need to be corrected.
Italics and standardized form need to be distinguished, especially formulas, parameters and units. - The article has been completed and corrected
The text of Fig 17 should be in English. – THE DRAWING DESCRIPTION HAS BEEN CHANGED.

Reviewer 4 Report
Comments and Suggestions for Authors
This paper introduces the use of stray magnetic field (MFL) to diagnose damage occurring on the lift guide rails. FEA simulation and experimental tests have been done to verify the method. However, the authors must improve the description of the originality and novelty of their works. I propose to reconsider this manuscript for publication after major revision, as detailed below
1. The authors must add more reference.
2. The writing of abstract section must be improved. You should not introduce the background too much, and must add the main method and the innovative results.
3. The writing of introduction section must be improved, and it just likes a simple list of reference. Furthermore, you must add the description of your main work in this paper in the last paragraph of the Introduction.
4. In the figure 1, the unit of length must be given.
5. Lines 183 to 189 is not standardized.
6. The material properties of all the parts in the figure 5 must be provided. Furthermore, the boundary and load condition are not clearly described in the finite element modeling.
7. The first B1 and the second B2 grinding stage are not clearly described, resulting in the hard understanding of Figs.13 to 16.
8. The relationship between the simulation results and experimental results is not clear presented.
9. The writing of conclusion section is too simple, which can not reflect the article innovation.
Comments on the Quality of English LanguageThe English writing of the manuscript is too poor and need to be revised carefully.
Author Response
Dear Reviewer,
Thank you for your review of our article, which will significantly increase its attractiveness to the reader. As a result of your review, we have made the following changes to the content of the article:
This paper introduces the use of stray magnetic field (MFL) to diagnose damage occurring on the lift guide rails. FEA simulation and experimental tests have been done to verify the method. However, the authors must improve the description of the originality and novelty of their works. I propose to reconsider this manuscript for publication after major revision, as detailed below
- The authors must add more reference – THE INDEX OF PUBLICATIONS HAS BEEN ENLARGED.
- The writing of abstract section must be improved. You should not introduce the background too much, and must add the main method and the innovative results.- The article has been completed and corrected
- The writing of introduction section must be improved, and it just likes a simple list of reference. Furthermore, you must add the description of your main work in this paper in the last paragraph of the Introduction. - The article has been completed and corrected
- In the figure 1, the unit of length must be given. – We have added the description of units next to the drawing caption.
- The material properties of all the parts in the figure 5 must be provided. Furthermore, the boundary and load condition are not clearly described in the finite element modeling. The article has been completed and corrected.
- The first B1 and the second B2 grinding stage are not clearly described, resulting in the hard understanding of Figs.13 to 16. The article has been completed and corrected.
- The relationship between the simulation results and experimental results is not clear presented. The article has been completed and corrected.
- The writing of conclusion section is too simple, which can not reflect the article innovation. The article has been completed and corrected.

Round 2
Reviewer 1 Report
Comments and Suggestions for Authors
Dear Authors,
this is the second review of the article entitled "Use of stray magnetic field to diagnose damage to the lift guide rails system" by Pawel Lonkwic and co-authors. Unfortunately, in my opinion, the article was not significantly improved, and some of the comments were not taken into account by the Authors.
1) in the abstract, the scientific goal should be placed before the description of the Authors' research methodology, not at the end!!! Why did the authors not refer to the quantitative research results in the abstract (this was the comment in the first review)?
2) note 6 from the first review - Each description of scientific publications begins with "In the article...". The description of current research results and applications of stray magnetic field should be understandable and user-friendly (it should be one story, discussing detailed aspects of the topic related to the article). The comment was not taken into account, the description in Chapter I Introduction of each article begins with the words: In the publication, the article or in the article.
3) note 9 from the first review, it was not about the italic text but about the length of the dash and the use of formatting in accordance with the journal's template.
4) comment 10 from the first review was not included. The correct formatting of figure and table captions is below (view from journal template).
5) line 224 - where is the description of the markings in the figure? They should be in the drawing caption.
6) note 12 has not been taken into account, it is not about the italic text, but about writing the number and unit, it is 0.3mm and should be 0.3 mm, there is a space missing!!!!
7) note 14, I have not found anywhere an explanation regarding the similarity of the defects examined in the reviewed article and in article [8] from the first version of the manuscript. Please refer in detail whether this is the same defect, this is what it looks like in the figure.
8) note 15 from the first review: 15) Authors present their results but without any discussion supported by the literature. When the results are not discussed and conveniently supported by the open literature, questionable conclusions are obtained. Currently, the article looks more like a report from simulation and test than a scientific article. What does it mean "poczatek uszkodzenia" in Figure 17? – THIS EXPRESSION IN FIGURE 17 HAS BEEN TRANSLATED. THE RESULTS OBTAINED HAVE BEEN DISCUSSED IN MORE DETAIL
Discussion of the results should be made in relation to the literature. The Authors added a paragraph describing the research results in Chapter 5. Conclusions, but they did not refer these results to the results published in other publications. Moreover, comparison of results with generally available results and discussion should be in an earlier chapter and not in the Conclusions.
9) In its current form (after modifications by the Authors!!!), the article does not meet the editorial requirements of the journal. Before submitting an article to a journal, it is worth saving the manuscript in a PDF file to see whether the text has been edited correctly. An example of incorrect text editing below.
Reviewer
Comments on the Quality of English LanguageDear Authors,
I asked to change those fragments of the text that contain a personal form. Unfortunately, this comment was not taken into account. Examples:
Line 79 - "As pioneers in this domain, they presented..."
Line 80 - "They described..."
Line 94 - "Then they optimized"
and others, a high-class scientific article is written in an impersonal form.
Reviewer
Author Response
Dear Reviewer,
Thank you very much for the sent comments of our article, which will certainly improve it. Below are the responses to the included comments on both the qualitative and editorial side of our article.
Best regards
Poul Lonkwic

Reviewer 3 Report
Comments and Suggestions for Authors
There is no comments and comments. I think that the manuscript can be accept in present form.
Author Response

(The authors gave the same response as above.)

Reviewer 4 Report
Comments and Suggestions for Authors
The paper has been revised and improved according to the suggestion, and I suggest it for publication.
Comments on the Quality of English LanguageThe English writing of the manuscript has been improved.
Author Response

(The authors gave the same response as above.)

Round 3
Reviewer 1 Report
Comments and Suggestions for Authors
Dear Authors,
the article has been improved and can be published in the Sustainability journal.
Reviewer